# Hormonal Masculinization of the European Grayling (*Thymallus thymallus*) Using 11β-Hydroxyandrostenedione (OHA) and 17α-Methyltestosterone (MT)

**DOI:** 10.3390/ani15203059

**Published:** 2025-10-21

**Authors:** Rafał Rożyński, Marcin Kuciński, Stefan Dobosz, Anna Kycko, Konrad Ocalewicz

**Affiliations:** 1Department of Marine Biology and Biotechnology, Faculty of Oceanography and Geography, University of Gdansk, Al. M. Piłsudskiego 46, 81-378 Gdynia, Poland; marcin.kucinski@ug.edu.pl (M.K.); konrad.ocalewicz@ug.edu.pl (K.O.); 2Department of Salmonid Research, National Inland Fisheries Research Institute in Olsztyn, Rutki, 83-330 Żukowo, Poland; s.dobosz@infish.com.pl; 3Department of Research Support, National Veterinary Research Institute in Puławy, Al. Partyzantów 57, 24-100 Puławy, Poland; anna.kycko@piwet.pulawy.pl

**Keywords:** neo-males, sex reversal, androgen exposure, gonadal differentiation, sterility, intersex

## Abstract

The European grayling is an important salmonid species whose natural populations have declined due to anthropogenic pressure on riverine ecosystems. Conservation efforts that involve stocking open waters with hatchery-reared fish can lead to genetic pollution in native grayling populations. Production of sterile triploid females might offer a solution to this problem. To produce such stocks, efficient masculinization methods for the species are needed to provide neo-males. This study preliminary assessed the potential of 11β-hydroxyandrostenedione (OHA) and 17α-methyltestosterone (MT) for masculinization of the grayling. OHA only induced the development of external male traits in up to 77% of fish, while MT frequently caused intersex differentiation. These findings indicate that hormonal masculinization in the grayling is feasible but requires further studies focused on dosage, timing and administration methods.

## 1. Introduction

The European grayling (*Thymallus thymallus*), hereafter referred to as the grayling, is a freshwater salmonid fish (order Salmoniformes, family Salmonidae, subfamily Thymallinae) native to European running waters. The grayling plays an important ecological role as a bioindicator of river ecosystem health, being particularly sensitive to elevated water temperature (>20 °C), low dissolved oxygen (<6 mg/L) and increased loads of nutrients and pollutants such as nitrates and heavy metals [1,2,3]. The grayling is also a highly valued species for recreational angling, especially in rivers in Central and Northern Europe, where thousands of licensed fly-fishers participate annually in mostly catch-and-release fisheries, contributing substantial local economic benefits through tourism and fishing license revenues [4,5].

Unfortunately, over recent decades, habitat degradation, predation pressure and excessive fishing have led to a marked decline in grayling abundance and distribution across Europe [6,7,8]. Specifically, electric-fishing surveys in southern Belgium showed a ~42.8% decline in the individual grayling abundance and ~52.6% loss in its biomass in 2000–2022, while in Bavarian streams, the number of grayling individuals dropped by about 41.7% between the early 1990s and mid-2000s [3]. To counteract this decline, conservation management measures, such as size limits (from 28 to 45 cm, depending on the country), closed seasons for spawning periods, catch quotas and restocking initiatives using hatchery-reared individuals from regional broodstock, have been implemented in recent years [1,5,9,10].

Multiple observations have revealed that the widespread introduction of graylings from non-native populations and hatchery-raised specimens raises a serious threat to the genetic integrity of many local populations of this species [7,11,12]. In several cases, the resulting loss of genetic identity has led to the complete disappearance of many local grayling populations due to reduced fitness [7,11,13]. To address this issue, the use of functionally sterile fish for stocking has been proposed as a strategy to prevent unwanted gene flow into wild populations.

In salmonid fish species, triploid females typically exhibit impaired gametogenesis, which makes them safe for wild fish populations if stocked [14]. Triploid all-female stocks can be obtained through a three-step process, including (1) the generation of all-female diploids via gynogenesis, (2) the production of neo-males (XX) through hormonal sex reversal of the genetic females, and (3) the triploidization of eggs fertilized with the sperm from the neo-males. While gynogenesis and triploidization protocols have already been developed for graylings [15,16], methods enabling the reliable production of grayling neo-males are still missing.

Hormonal masculinization in salmonids is usually induced through the treatment of the fish with androgens such as 11β-hydroxyandrostenedione (OHA) or 17α-methyltestosterone (MT) [17,18,19,20,21]. These compounds act by binding to androgen receptors and inhibiting aromatase, thereby redirecting undifferentiated gonads toward testicular development [22,23]. Effective masculinization depends on the correct hormone dose and application during the critical developmental window prior to gonadal differentiation [24,25].

In this study, the potential of OHA and MT for the masculinization of the grayling was assessed to provide preliminary data supporting the development of a reliable protocol enabling the production of neo-males.

## 2. Materials and Methods

### 2.1. Fish Stock Origin and Maintenance

The experiments were performed using graylings from the broodstock kept at the Department of Salmonid Research (DSR), National Inland Fisheries Research Institute, in Olsztyn, Poland (Rutki; 54°19′51.1″ N 18°20′15.1″ E). Graylings from the broodstock used in the research reached sexual maturity in their second year of life. The spawning fish, aged three to four years, were raised in a 3000 m^3^ flow-through pond system supplied continuously with water from the Radunia River at a rate of 5 L/s. The inflowing river water exhibited a seasonal temperature range of 2–17 °C, with daily fluctuations not exceeding ±0.5 °C and dissolved oxygen levels consistently above 80% saturation (8.5–11.5 mg/L) throughout the year. They were fed a commercial broodstock diet (Aller Bronze 3 mm and Aller Silver 3 mm, Aller Aqua, Golub-Dobrzyń, Poland) three to four times daily, with feed amounts ranging from 0.5% to 1.5% of their biomass depending on water temperature. Feeding was stopped two weeks before spawning, and experimental crossings took place on 29 April 2021 and 24 April 2023.

Gametes were collected from ten females and five males, which measured 26.3–35.5 cm in body length and 141–393 g in body weight. This female/male ratio is commonly used in aquaculture to ensure a sufficient number of eggs, while keeping the number of males manageable. Before gamete collection, fish selected for stripping were anesthetized with MS-222 (50 mg/L, Sigma-Aldrich, Madrid, Spain). Eggs stripped from all females were pooled and thoroughly mixed before fertilization with pooled milt. Sperm quality was evaluated using a computer-assisted sperm analysis (CASA) system after activation in sperm activation medium (SAM; 1 mM CaCl_2_, 20 mM Tris, 30 mM glycine, 125 mM NaCl, pH 9.0). The quantitative parameters measured included total motility (%), progressive motility (%), curvilinear velocity (VCL, µm/s), straight-line velocity (VSL, µm/s), and average path velocity (VAP, µm/s), recorded 10 s post-activation. Only ejaculates exhibiting ≥80% total motility were used for fertilization.

Eggs were incubated at a temperature of 7.0 ± 0.5 °C, with oxygen levels maintained at 10.8 ± 0.5 mg/L and pH at 7.5 ± 0.1, in separate egg trays in vertical incubators under the salmonid hatchery’s routine sanitary conditions [26]. The system was continuously supplied with fresh, oxygenated river water, and metabolic waste (ammonia and nitrite) levels were monitored weekly using colorimetric tests (JBL GmbH, Neuhofen, Germany), remaining below 0.05 mg/L for NH_3_ and 0.01 mg/L for NO_2_^−^ throughout incubation. Following complete yolk sac resorption (at 70°D post-hatching, i.e., 70 accumulated degree-days calculated as the product of days and water temperature in °C), the grayling hatchlings were fed *Artemia nauplii* for 10 days. In this species, the active swim-up fry are large enough, with sufficiently developed jaws, to be directly fed on Artemia, as commonly practiced in aquaculture.

The produced fry were used to conduct two experiments that focused on the hormonal sex reversion and production of neo-males (genetic females). For this purpose, 11β-hydroxyandrostenedione (OHA) (Sigma-Aldrich, Madrid, Spain) was used in experiment 1 and 17α- methyltestosterone (MT) (Sigma-Aldrich, Madrid, Spain) was used in experiment 2. Both hormones were administered in variable doses by feeding the fish with specially prepared hormone-enriched feed.

### 2.2. Diet Preparation

Two types of commercial feed were used depending on the fish size, namely Aller Aqua Infa EX GR 0.4 mm for fish up to 2 g (approximately 15–50 mm in body length) and Aller Aqua Futura EX GR 0.5–1.0 mm for fish of 2–5 g (approximately 50–80 mm in body length) (Aller Aqua, Golub-Dobrzyń, Poland). For masculinization experiment 1, OHA feed with doses of 10 mg (OHA_10ppm_) and 20 mg (OHA_20ppm_) per kg was used. For experiment 2, MT feed with doses of 3 mg (MT_3ppm_) and 6 mg (MT_6ppm_) was utilized. For both experiments, non-enriched feeds were also used as the control diets (OHA_C_ and MT_C_). Hormone supplementation of the feeds was carried out using the sprinkle method. For this purpose, each hormone supplied as dry powder (Sigma Aldrich, Madrid, Spain) was accurately weighed, dissolved in ethanol and evenly applied to the commercial feed by hand spraying. The feeds were then dried at room temperature for 24 h to ensure complete ethanol evaporation and subsequently stored at 4 °C until use. To eliminate the potential effect of ethanol on the feed palatability, the control diet underwent the same spraying process but without the addition of the active compounds.

### 2.3. Experimental Design

The hormonal sex reversal experiments were carried out on 18 June 2021 (experiment 1) and on 13 June 2023 (experiment 2) using grayling fry 20 days post-hatching, weighing, on average, 82–85 mg, with fully resorbed yolk sacs and exhibiting active feeding behavior. For each experiment, a total of 450 fish were divided into three equal groups (150 indiv. per group) and then stored in nine separate 300 L flow-through tanks (1 L/s water flow), with 50 fish per tank, corresponding to three treatments (two hormonal doses and one control), each replicated in triplicate. Each setup included two experimental tanks, i.e., OHA_10ppm_ and OHA_20ppm_ (for experiment 1) or MT_3ppm_ and MT_6ppm_ (for experiment 2), along with control tanks, i.e., OHA_C_ and MT_C_, respectively. Although immersion treatments can be used in salmonids, in this study we focused on oral administration throughout the critical period of gonadal differentiation to achieve continuous hormone exposure, ensuring sufficient uptake while minimizing handling stress.

The hormone administration phase lasted 80 days (experiment 2) and 83 days (experiment 1). Further, the fish were fed with Aller Aqua Infa EX GR 0.4 mm, until their average weight exceeded 5 g per individual. After this period, the fish were transferred to larger tanks with a capacity of 1200 L and a water flow of 4 L/s, where the second phase of each experiment commenced. For the next 22 (experiment 1) and 12 (experiment 2) months, fish were fed Aller Futura GR 0.5–2.0 feed without hormone supplementation. The fish were hand-fed eight times per day, with the daily feed ration reduced by 20% compared to the recommended amount for rainbow trout at a specific temperature, following From’s and Rasmussen’s [27] guidelines. During the hormone administration phase, the feed consumption was strictly monitored to ensure that >95% of the rations were taken. Feed was replaced twice daily to prevent hormone degradation and ensure continuous availability.

### 2.4. Assessment of Masculinization Rate

Masculinization effectiveness in fish from experiment 1 was assessed by analyzing dimorphic traits and gonadal morphology, combined with PCR-based molecular genetic sex verification. Taking into account the fact that a reliable assessment of the gonadal differentiation and masculinization effect in graylings from the Rutki broodstock may be performed as early as 24 months after hatching, 25-month-old fish were examined, including 66 and 77 individuals from the experimental groups (OHA_10ppm_ and OHA_20ppm_, respectively) and 55 individuals from the control group (OHA_C_). The unequal group sizes reflect natural survival variability during the 22-month rearing period, and all surviving individuals from each group were included to maximize statistical power and representativeness. In turn, masculinization efficacy in fish from experiment 2 was assessed using PCR-based molecular genetic sex identification and gonad histology. For molecular analysis, 20 randomly selected 15-month-old individuals each from the experimental (MT_3ppm_, MT_6ppm_) and control (MT_C_) groups were analyzed. Gonadal histology was then performed on seven genetically verified females from each experimental group (MT_3ppm_, MT_6ppm_) and six from the control group (MT_C_). All fish were euthanized using an overdose of MS-222 (50 mg/L) and dissected to collect the gonads for morphological or histological observations. For PCR-based molecular genetic sex verification, small pelvic or pectoral fin clips were collected from each fish, preserved in 96% ethanol and stored at a temperature of 4 °C until genetic material extraction.

The external sexual dimorphism analysis was performed following the procedure outlined by Witkowski [28]. Fish were classified as males if they displayed breeding tubercles on the posterior part of the body, dark body coloration, a large cherry-colored dorsal fin, and a small genital papilla. Females were identified by a small dorsal fin, light body coloration, and a large genital papilla. In turn, fish exhibiting a silvery body and lacking a genital papilla were classified as indeterminate. The external sexual dimorphism analysis was confirmed through the morphological observation of the dissected gonads. Males were identified by the presence of paired, elongated testes with a milky or whitish appearance, females by paired, lobulated ovaries containing visible oocytes, and sterile individuals by rudimentary, thread-like gonads lacking differentiated gametes. For data collection, all fish and their gonads were photographed with a Canon G16 camera (Canon, Tokyo, Japan). Each image was visually inspected to identify and characterize every distinct sex identified among the examined fish. Images were captured on a neutral gray background with consistent LED lighting to avoid shadows, and a scale bar was included in each frame. Camera settings were standardized with ISO 200, aperture f/5.6 and magnification adjusted to capture the entire specimen or gonad. No automated image analysis software was used. All images were visually inspected to identify and characterize each distinct external sexually dimorphic phenotype among the examined fish.

For PCR-based molecular genetic sex verification, genomic DNA was isolated from collected and preserved fin clips using the standard Chelex-100 method [29]. The quality and amount of the isolated genomic DNA were checked with the NanoDrop™ One spectrophotometer (Thermo Scientific, Waltham, MA, USA). The genetic sex of the sampled fish was verified through the PCR duplex amplification of the Y-chromosome-linked DNA marker (sdY) (F: ATGGCTGACAGAGAGGCCAGAATCCAA, R: CTGTTGAAGAGCATCACAGGGTC) and 18S rDNA (F: GTYCGAAGACGATCAGATACCGT, R: CCGCATAACTAGTTAGCATGCCG) as a positive control [30]. The PCR amplifications were carried out using 10 ng of the isolated DNA template in a reaction mixture of 12.5 μL total volume composed of 1× PCR Master Mix Green Plus (A&A Biotechnology, Gdańsk, Poland), as well as 0.4 μM of each SdY and 0.1 μM of each 18S rDNA primer. PCR amplifications were performed with a Mastercycler^®^ X50s (Eppendorf, Hamburg, Germany) under the following conditions: initial denaturation at 96 °C for 4 min, followed by 35 cycles at 94 °C for 30 s, annealing at 60 °C for 45 s, elongation at 72 °C for 45 s and a final elongation step at 72 °C for 10 min. The resulting products of the PCR amplifications were separated in a 2.0% agarose gel (Sigma-Aldrich, St. Louis, MI, USA), stained with ethidium bromide (0.05 mg/mL) and then visualized by a UV trans-illuminator (Vilber Laurmat ECX-20.M, Eberhardzell, Germany).

For gonadal histology, the sampled tissues were immediately fixed in Bouin’s fluid after collection and stored at room temperature until further processing. The preserved gonad tissues were subsequently dehydrated in a series of increasing concentrations of ethanol, cleared in xylene, and embedded in paraffin blocks using the tissue processor Leica TP 1020 (Leica Biosystems, Park, IL, USA) and a paraffin dispenser (DP500, Bio-Optica, Milano, Italy). Then, 5 μm thick sections were cut from each gonadal tissue using a Microm HM 350S microtome (Microm, Walldorf, Germany). The slides were next stained using a routine hematoxylin–eosin (HE) staining method [31,32]. Histological analysis was performed using an Axiolab light microscope (Zeiss, Oberkochen, Germany), with an Axiocam 208 color camera and Zen 3.4 software (Zeiss, Oberkochen, Germany). The classification of germ cells and gonadal cellular structures followed the nomenclature of Król et al. [33], with gonads categorized as follows:(A)Testes—spermatogonia type A;(B)Testes—spermatogonia type A and spermatozoa;(C)Ovaries—oogonia and oocyte type I and II;(D)Ovaries—oogonia and oocyte type I, II and III;(E)Ovaries—oogonia, oocyte type I and II, and single spermatogonia;(F)Ovaries—oogonia, oocyte type I, II, and III, single spermatogonia, and spermatozoa.

### 2.5. Statistical Analysis

For experiment 1, the proportions of external sexual dimorphism phenotypes (male-like, female-like, and sterile-like individuals) between the experimental (OHA_10ppm_ and OHA_20ppm_) and control (OHA_C_) groups were compared using the chi-square (χ^2^) test of independence. For post hoc pairwise comparisons, adjusted residuals were examined to identify significant deviations from expected frequencies. In experiment 2, gonadal histology for all groups (MT_3ppm_, MT_6ppm_ and MT_C_) was analyzed descriptively, without statistical testing due to the small sample size, as this experiment was designed as a preliminary study. A significance threshold of *p* < 0.05 was applied for all statistical analyses. Proportional 95% confidence intervals were calculated using the Wilson score method. All calculations were performed using Statistica 13.3 (TIBCO Software Inc., Palo Alto, CA, USA).

## 3. Results

### 3.1. Masculinization Rate in Experiment 1

The effectiveness of masculinization in fish fed 11β-hydroxyandrostenedione (OHA) was assessed using external sexual dimorphism, gonadal morphology, and PCR-based molecular genetic sex verification. Examination of external sexual dimorphism revealed fish displaying male-like, sterile-like, and female-like external dimorphism in the examined graylings (Figure 1). The highest percentage (76.6%) of fish displaying external male dimorphism was observed in the OHA_20ppm_ group (59 indiv.) (Figure 1A and Figure 2). In the OHA_10ppm_ group, this percentage was 66.7% (44 indiv.), while in the OHA_C_ group, it was 54.5% (30 indiv.). The highest share (12.7%) of fish classified as sterile was recorded in the OHA_C_ group (seven indiv.), while in the OHA_10ppm_ and OHA_20ppm_ experimental groups it varied from 7.6% (five indiv.) to 7.8% (six indiv.), respectively (Figure 1B and Figure 2).

Examination of gonadal morphology revealed that fish from both OHA groups and the control group classified based on external sexual dimorphism as females and sterile specimens had developed ovaries (Figure 1B,C). Among fish displaying external male dimorphism, examination of gonadal morphology revealed that 2 individuals (4.5%) in the OHA_10ppm_ group and 17 individuals (28.8%) in the OHA_20ppm_ group exhibited ovaries, while the remaining fish showed testes (Figure 1A). Genetic analysis confirmed that all fish exhibiting ovarian development were genetic females, while those with testicular development were confirmed to be genetic males.

### 3.2. Masculinization Rate in Experiment 2

In Experiment 2, the effectiveness of sex reversal using 17α-methyltestosterone (MT) was assessed through PCR-based molecular genetic sex verification and gonad histology. Among fish that were molecularly verified as females, five specimens (71.4%) from the MT_3ppm_ group and three specimens (42.9%) from the MT_6ppm_ group were found to have properly developed ovaries, corresponding to gonad types C and D (Table 1 and Figure 3). Intersex gonads (types E and F: ovaries with oogonia, oocytes stages I–III and scattered spermatogonia together with spermatozoa) were observed in two genetic females (28.6%) from the MT_3ppm_ group and in four molecularly confirmed females (57.1%) from the MT_6ppm_ experimental group. In the control group, both genetic males and genetic females developed well-formed testes and ovaries, respectively (gonad types A–D), consistent with their genetic sex.

## 4. Discussion

In fishes, sex differentiation is largely controlled by steroids and gonadotropins, produced primarily by the brain during early development and later also by the gonads [23]. In the Salmoninae family, most species exhibit differentiated gonochorism, in which early development of gonads continues from the indifferent gonad directly to the testes or ovaries. This mode of gonad development has been documented in Arctic charr (*Salvelinus alpinus*), brook charr (*S. fontinalis*), brown trout (*Salmo trutta*), rainbow trout (*Oncorhynchus mykiss*) and coho salmon (*O. kisutch*) [34,35,36,37,38]. In contrast, representatives of the Thymallinae subfamily display undifferentiated gonochorism, where gonads initially develop along a single pathway—usually female-like—before later differentiating into testes or ovaries [23]. The European grayling represents a rare variant within this pattern, as its gonads first exhibit an all-male-like stage around the 7th week post-hatching before transitioning to ovarian differentiation from approximately the 11th week post-hatch prior to developing mature testes and ovaries [39].

The main goal of the present study was to evaluate the potential use of 11β-hydroxyandrostenedione (OHA) and 17α-methyltestosterone (MT) for masculinization in the grayling. MT is a synthetic androgen widely used in aquaculture to induce sex reversal by mimicking the action of endogenous testosterone and thus stimulating testicular differentiation during the labile period of gonadal development [23,40]. OHA, in turn, is a natural metabolite of testosterone and a potent androgen precursor, which can be enzymatically converted into active androgens within fish tissues, primarily via the 17β-hydroxysteroid dehydrogenase and 5α-reductase enzymes, thereby promoting male gonadal development [41,42,43]. Both OHA and MT masculinize fish primarily by activating androgen receptors in the developing gonads, while simultaneously suppressing estrogen synthesis through competitive inhibition of aromatase activity [22]. Because MT often impairs gonadal development, it is frequently replaced by OHA, which effectively induces masculinization with fewer intersex cases, less sterility, and reduced adverse effects [17,18,20].

In this study, the dietary administration of OHA slightly increased the ratio of individuals displaying male secondary sexual characteristics in graylings, including body coloration, fin shape and genital papilla, but it did not affect gonadal development. This contrasts with findings in rainbow trout, where oral OHA administration to the swim-up fry for up to 57–60 days at doses comparable to those used in the present study (10–20 ppm) resulted in masculinization rates that varied from 80 to 100% [18,20]. In turn, feeding rainbow trout fry with OHA-supplemented diets at higher doses (60 ppm) for 20–60 days after yolk sac resorption resulted in approximately 70% sex-reversed neo-males and 20% intersex individuals [19]. On the other hand, lower doses of this hormone (5 ppm) provided for 57 days from the onset of exogenous feeding resulted in a reduced masculinization rate (about 30%) [18]. Treatment of the European whitefish (*Coregonus lavaretus*) at 32 days post-hatching with feed enriched with OHA at doses of 10 and 20 ppm for 63 days resulted in gonadal sterilization and the appearance of intersexual gonads [21]. The administration of OHA through immersion (0.4 ppm for 2 h) and oral supplementation (3 ppm for 60 days) one to three times between hatching and first feeding resulted in approximately 70% sex-reversed neo-males in rainbow trout [17]. Similarly, Redding et al. [44] showed that the immersion of swim-up fry in an OHA solution (0.5 ppm for 4 h, three times per week), combined with dietary administration of OHA at doses of 5–50 ppm for 4–8 weeks, induced testis-like gonadal development in 98% of coho salmon (*Oncorhynchus kisutch*) and chum salmon (*O. keta*) individuals.

The administration of MT induced dose-dependent disruptions in gonadal development in the grayling genetic females, resulting in intersex gonads rather than developed functional testes, which suggests that in graylings MT is a more masculinizing agent than OHA. In rainbow trout, feeding fry with diets supplemented with MT (5 ppm) for 57 days from the onset of exogenous feeding resulted in the masculinization of 40% of females, while the remaining 60% of individuals developed intersex gonads [18]. A slightly higher dose of MT (6 ppm) administered in feed for 60 days to all-female rainbow trout fry resulted in approximately 60% sex-reversed neo-males and only around 2% intersex individuals [20]. In turn, Atar et al. [19] reported a sex reversal rate of approximately 90% in this species after 60 days of feeding with diets containing 3ppm of MT hormone. In contrast, feeding mixed-sex fry with higher MT doses (20 ppm) resulted in 50% sterilization, with 40% males and 10% intersex individuals [18]. A single 2 h immersion of fry in 0.4 ppm of MT before the onset of exogenous feeding (1–2 weeks post-hatch), followed by 60 days of feeding with 3 ppm of MT, resulted in the complete masculinization of rainbow trout [17].

Baker et al. [45], studying chinook salmon, achieved 82% males following fry immersion in 2–20 ppm of MT and concluded that the efficiency of hormone-induced masculinization in salmonids strongly depends on species-specific sensitivity, as well as the dose, route and timing of administration. In general, excessive doses or prolonged exposure typically cause complete sterilization, whereas insufficient doses often result in no sex reversal or the development of intersex gonads. While exact dose thresholds for graylings remain to be established, studies in other salmonids indicate that, for 17α-methyltestosterone, dietary doses above ~20 ppm or prolonged exposure during the critical labile period can induce sterilization, whereas doses of 3–6 ppm often result in partial masculinization with frequent intersex occurrence [18,20]. It is hypothesized that the aromatization of MT or endogenous testosterone to estradiol in the gonads leads to intersex development, particularly when the administered hormone dose is too low [46,47].

Recent studies have demonstrated that sex-specific gene expression in the grayling can already be detected during embryogenesis and is particularly pronounced around hatching, but gonadal differentiation is strongly delayed [39]. At hatching and for at least three weeks thereafter (22 days post-hatching, dph), gonads remain undifferentiated, containing only a few primordial germ cells. The first signs of differentiation appear in the grayling in around the 7th week post-hatching (~50 dph), when immature testis-like structures are present in about 50% of juveniles, most of which are genetic females. This male-like phenotype in females is progressively replaced by ovarian development from about the 11th week post-hatch (~78 dph) [39]. In contrast, males often retain undifferentiated gonads for much longer, with only ~12% showing testicular tissue in the 18th week post-hatch (~130–134 dph). However, final gonadal differentiation in both sexes occurs no earlier than 29 weeks post-hatching (~208–216 dph) [39]. Taken together, these findings indicate that the period of androgen sensitivity in graylings is largely protracted and delayed, spanning well beyond the onset of exogenous feeding, i.e., from the peri-hatch period to approximately eight months post-fertilization. This suggests that the recorded discrepancy of external masculinization without gonadal reversal in the examined graylings can likely be attributed to the hormone administration period being too early and short. Therefore, hormonal treatment to produce grayling neo-males may require a later onset and/or an extended duration, potentially combined with immersion and oral administration. However, it cannot be discounted that the metabolic fate of OHA and MT hormones in graylings may differ from that in other salmonids, as they can be preferentially metabolized to weaker androgens or rapidly inactivated. As a result, the effect of both hormones may be limited to tissues that are sensitive to circulating androgens (skin, fins, secondary sexual traits), without sufficient androgenic stimulation of the gonads. A similar dissociation between external and gonadal masculinization has been described in cases where aromatase activity remains high, leading to persistent estrogen synthesis that counteracts testicular differentiation [46,47].

## 5. Conclusions

To conclude, the obtained results indicate that androgen-mediated neo-male induction by OHA and MT hormones is potentially possible in the grayling, but it requires substantial refinement. In particular, the efficient masculinization of this species by hormonal treatment may require higher doses of hormones, different ways of administration (a combination of immersion and oral treatment), earlier (before hatching) treatment, and prolonged exposure (up to 29 weeks post-hatch).

## Figures and Tables

**Figure 1 animals-15-03059-f001:**
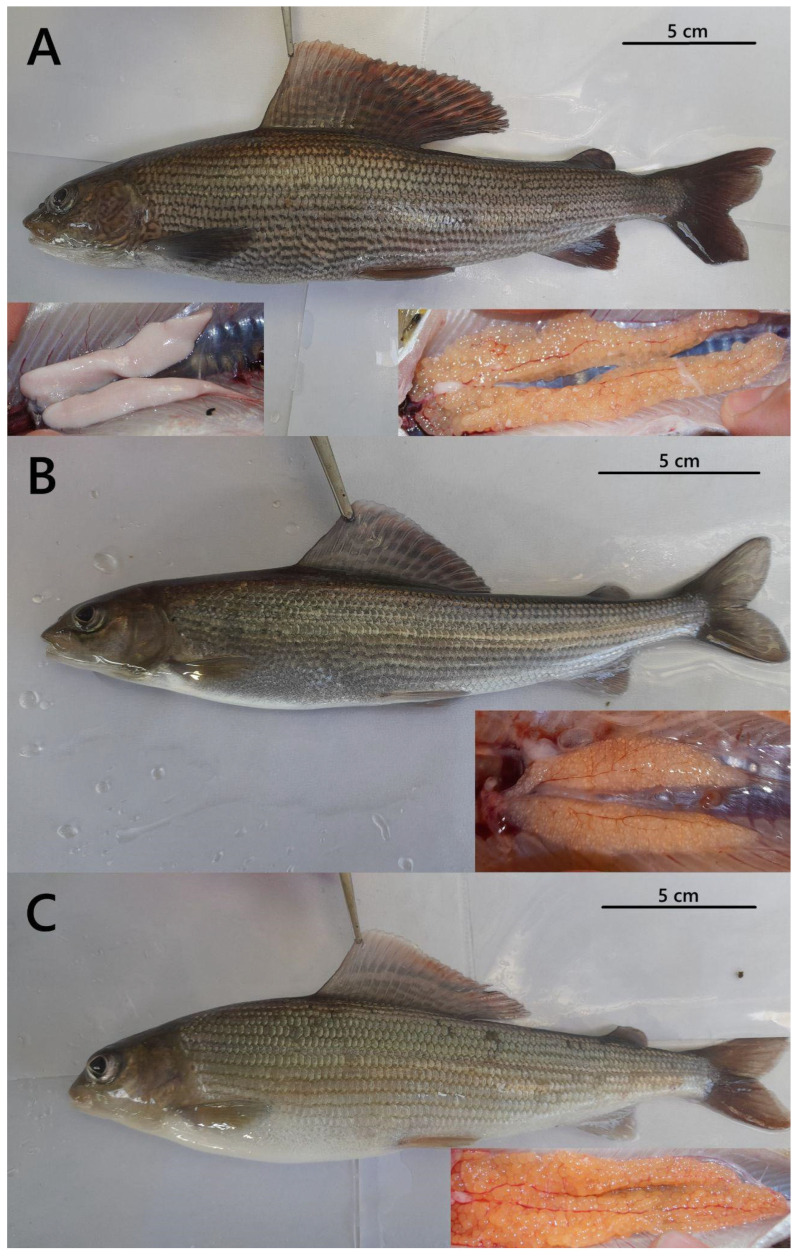
Types of recorded external sexual dimorphism along with gonad morphology observed in graylings fed 11β-hydroxyandrostenedione-enriched (OHA) feed: (**A**) male-like external dimorphism: fish with breeding tubercles, dark body coloration, a large cherry-colored dorsal fin and a small genital papilla; (**B**) sterile-like phenotype: fish showing a silvery body and lacking a genital papilla; and (**C**) female-like external dimorphism: fish with light body coloration, a small dorsal fin and a large genital papilla. All specimens ranged from 23.2 to 27.9 cm in total length and from 107 to 150 g in body mass.

**Figure 2 animals-15-03059-f002:**
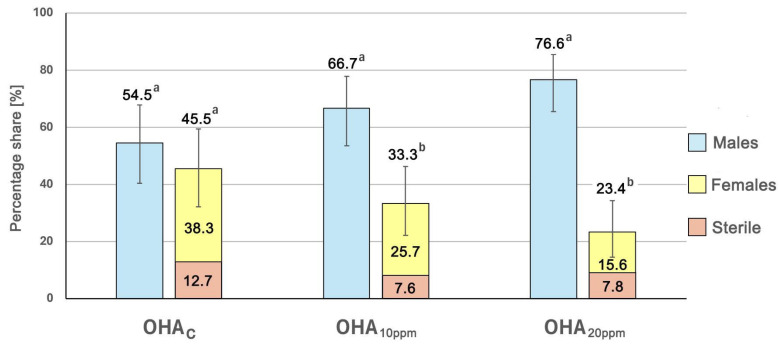
Percentage of fish displaying external sexual dimorphism in both experimental groups (OHA_10ppm,_ OHA_20ppm_) and the control group (OHA_c_). Different letters denote statistically significant differences (*p* < 0.05) in the proportions of external dimorphism phenotypes within groups. Error bars represent 95% confidence intervals.

**Figure 3 animals-15-03059-f003:**
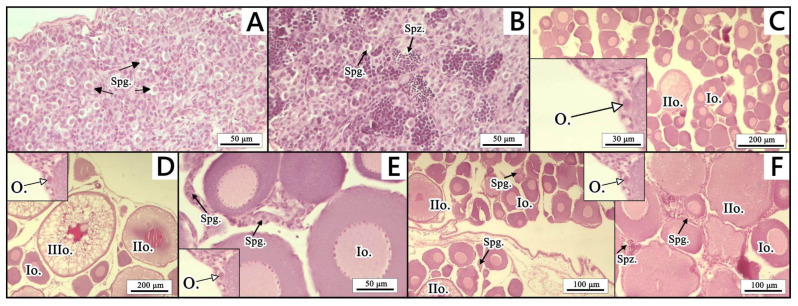
Gonadal development in the control group (MT_C_) and two experimental groups (MT_3ppm_ and MT_6ppm_) based on histological sections stained with hematoxylin and eosin (H&E). (**A**) Testes—spermatogonia type A; (**B**) testes—spermatogonia type A and spermatozoa; (**C**) ovaries—oogonia and oocyte type I and II; (**D**) ovaries—oogonia and oocyte type I, II and III; (**E**) ovaries—oogonia, oocyte type I and II, and single spermatogonia; (**F**) ovaries—oogonia, oocyte type I, II, and III, single spermatogonia, and spermatozoa. (Spg.—spermatogonia type A; Spz.—spermatozoa; O.—oogonia; Io.—oocyte type I; IIo.—oocyte type II; IIIo.—oocyte type III).

**Table 1 animals-15-03059-t001:** The percentage of grayling individuals with different gonad types in the control group (MT_C_; both sexes) and in the two experimental groups (MT_3ppm_ and MT_6ppm_; only individuals molecularly verified as genetic females).

Types of Gonads	MT_C_ (*n* = 6)	MT_3ppm_ (*n* = 7)	MT_6ppm_ (*n* = 7)
(A) Testes—spermatogonia type A	33.33%	-	-
(B) Testes—spermatogonia type A and spermatozoa	33.33%	-	-
(C) Ovaries—oogonia and oocyte type I and II	16.67%	57.13%	28.57%
(D) Ovaries—oogonia and oocyte type I, II and III	16.67%	14.29%	14.29%
(E) Ovaries—oogonia, oocyte type I and II, and single spermatogonia	-	14.29%	28.57%
(F) Ovaries—oogonia, oocyte type I, II and III, single spermatogonia, and spermatozoa	-	14.29%	28.57%

## Data Availability

The data in the present manuscript are available from the corresponding author upon request.

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
