# Peer review of "Hormonal Masculinization of the European Grayling (Thymallus thymallus) Using 11β-Hydroxyandrostenedione (OHA) and 17α-Methyltestosterone (MT)"

_animals, 2025, doi:10.3390/ani15203059_

Round 1
Reviewer 1 Report
Comments and Suggestions for Authors
A brief summary:
This MS by Abdallah et al. investigates the Pilot trials of hormonal masculinization conducted by feeding 20-day post-hatch fry with diets supplemented with OHA or MT for ~80 days. In the OHA-treated groups, the proportion of individuals showing external male-like dimorphism ranged from 66.3% (OHA10ppm) to 76.9% (OHA20ppm), with 3.8% (OHA10ppm) and 28.6% (OHA20ppm) developing ovaries consistent with genetic sex. In the MT-treated groups, intersex gonads were observed in 28.7% (MT3ppm) and 57.1% (MT6ppm) 41 of genetic females. The results indicate that androgen-mediated neo-male induction by OHA and MT is possible in the species but requires optimization of dose, timing and delivery, potentially combining embryonic immersion with prolonged dietary administration during the first 6–8 months of development.
- General concept comments.
This article is “classical” in term of goal, methods and results. I think there is something new in term of methods applied and type of results gotten according to the authors, this is the first time that such data were obtained on this species.
Overall, I think this is a good study in a system with limited previous knowledge of this level on this specific topic. I appreciated their approach and the time course involved in this work and think it add significant merit to their work. The Materials and Methods are clear and their overall conclusions were related to results. I think this work should undergo major revision before acceptance. I also have a couple suggestions.
- Specific comments
Line 83-84 “OHA” and “MT” had appeared in Line 75-76. There is no need to repeat the full name again.
Line 91 How about the body length and weight of the spawning fish?
Line 92 “5 dm³/s” change to “5 L/s”, better to use the international unit.
Line 98 How about the body length and weight of ten females and five males used in this experiment?
Line 99 “50 mg/dm³” change to “50 mg/ L”
Line 101 How to evaluate the sperm quality? What are the standards? Make it clear.
Line 107 “70°D post-hatching” What does it mean here?
Line 107 The larvae should be fed with small and large rotifers first, followed by Artemia nauplii. It should be clearly explained.
Line 116-117 How about the body length for fish up to 2g or 2-5 g?
Line 134 “300 dm³” change to “300 L”
Line 140 “1200 dm³” change to “1200 L”
Line 140 4 l/s What does it mean? Is it 4 L/s?
Line 146 “2.4. Verification of Masculinization Effectiveness” In the results, there is no any information on Masculinization effectiveness information? Are these results related Fig 2 for dimorphic traits? I suggest to make it clear in the results.
Line 179 What is the primer sequence of SdY and 18S rDNA used for PCR amplifications?
Line 216 In each grouper, how many fish was used in this experiment? It is best to mark it on the graph or explain it in the results. The result is not only about proportion, but also about the number of fish to make it clear.
Line 218-225 The author should explain and label Fig 2A, Fig 2B and Fig 2C in the explanation on figure 2, not only Fig. 2 at the end. What are the weight and length of the fish in figure 2?
Line 227-229 A scale bar should be added to each image in figure 2.
Line 242-246 The small image (O) inserted in the picture is not very clear. Make it clear.
Line 318 “days post-hatch” change to “days post-hatching”
The discussion was prepared well.
Reviewer 2 Report
Comments and Suggestions for Authors
Overall Evaluation: This manuscript investigates the use of 11β-hydroxyandrostenedione (OHA) and 17α-methyltestosterone (MT) for hormonal masculinization of European grayling (Thymallus thymallus) to produce neo-males for conservation purposes, such as generating triploid all-female populations. The topic is relevant to salmonid aquaculture and conservation biology, as wild grayling populations are declining, and the use of sterile stocking material can reduce genetic introgression risks. The study provides preliminary data on hormone effects, showing that OHA induces external male dimorphism in a dose-dependent manner, while MT often results in intersex gonads. However, the manuscript has significant methodological flaws, incomplete data analysis, and overinterpreted results. The experimental design lacks adequate controls and replicates, and the small sample size undermines the reliability of conclusions. While the work shows potential, substantial revisions are needed to enhance scientific rigor before it can be considered for publication.
Recommendation: Major Revision The manuscript requires major revisions to improve scientific rigor, expand the discussion of limitations, and provide more robust evidence. The authors should conduct additional analyses (e.g., statistical tests) and possibly supplementary experiments to validate findings. Resubmission is encouraged after addressing the issues below.
Major Issues:
- The introduction needs significant reorganization to better articulate the necessity of neo-male production for European grayling conservation and to comprehensively review masculinization protocols for related salmonid species. The current content is lengthy and lacks focus.
- The methods section lacks details on statistical analysis; please include appropriate statistical tests (e.g., chi-square test for sex ratio comparisons) and report p-values between treatment and control groups to enhance result credibility.
- The sample size for histological analysis in Experiment 2 is too small (n=7 per group), making robust conclusions difficult; increase sample size or provide power calculations to avoid Type II errors.
- The dietary hormone administration protocol needs data on feed intake rates and hormone stability, as exposure variability may confound results; consider combining immersion and extended oral administration.
- Percentage results lack confidence intervals; please add confidence intervals for all proportions (e.g., male dimorphism rate) to improve reliability and facilitate meta-analysis.
- Figure legends for Figures 1, 2, and 3 are incomplete, lacking scale bars or magnification details; revise all figures and tables, including details of histological staining methods.
- The discussion needs expansion to critically compare findings with prior OHA and MT studies in salmonids and explain why OHA induces only external masculinization without full gonadal reversal.
- The manuscript has language and grammatical issues (e.g., inconsistent tense, awkward phrasing such as “harm-ing native gene pools”); a thorough English edit by a native speaker is recommended.
- The reference list is outdated and incomplete; update with post-2020 literature on fish sex reversal and ensure all claims (e.g., triploid sterility) are properly cited to enhance scientific rigor.
- The abstract overemphasizes the feasibility of the method while failing to acknowledge limitations (e.g., MT’s intersex rate); revise to balance successes with the need for optimization.
Reviewer 3 Report
Comments and Suggestions for Authors
Review for the paper “Hormonal masculinization of European grayling (Thymallus thymallus) using 11β-hydroxyandrostenedione (OHA) and 17α-methyltestosterone (MT)” by RafaÅ‚ RożyÅ„ski and co-authors submitted to “Animals”.
The authors of this research paper conducted an analysis focused on the masculinization of European grayling, an ecologically significant salmonid species experiencing population declines due to various environmental stressors. The study specifically the effects of hormonal treatments with 11β-hydroxyandrostenedione (OHA) and 17α-methyltestosterone (MT) aimed at producing neo-males for large-scale production of triploid females. The results indicated a significant degree of success with both hormonal treatments: OHA showed promising outcomes in promoting male-like external characteristics, whereas MT resulted in intersex conditions in a notable proportion of genetic females.
The results of this study may have important implications for future conservation efforts aimed at restoring European grayling populations. By establishing a reliable technique for neo-male production, the findings could lead to the development of more ecologically sound stocking practices that maintain genetic diversity while supporting population recovery. Furthermore, this research opens avenues for optimizing dosage and timing in hormone administration, which may enhance masculinization efficiency.
Recommendations.
Introduction.
L 52-54. The authors should report which specific water-quality parameters (e.g., temperature, dissolved oxygen, pollutant loads) grayling abundance or physiology responds to. The authors should also explain the scale and economic value of fly-fishing for grayling (annual angler numbers, catch-and-release rates, regional hotspots) to support the claim of its recreational importance.
L 54-56. The authors should quantify the magnitude and time frame of declines (e.g., percent range contraction per decade).
L 57. The authors should specify which regulations (size-limits, closed seasons, catch quotas), restocking protocols (hatchery vs. wild broodstock, release densities).
Material and Methods.
L 89-92. The authors should report the daily and seasonal variability in temperature and dissolved-oxygen levels if available and describe whether this was a flow-through or recirculating system.
L 98-99. The authors should explain the rationale for the 2:1 female-to-male ratio.
L 101. The authors should report which quantitative sperm-quality metrics were assessed (percent motility, velocity parameters) and at what time point(s) post-activation measurements were taken.
L 104-106. The authors should describe the model and configuration of the incubators and how waste metabolites (ammonia, nitrite) were monitored and controlled.
L 133. The authors should clarify whether each hormonal dose and its control were replicated across multiple tanks (i.e., true biological replicates).
L 149-150. The authors should explain the rationale for selecting 25-month-old fish. They should also justify the unequal group sizes.
L 170-173. The authors should describe the photographic setup (lighting, scale bars, background), image-acquisition settings (ISO, aperture, magnification). They should clarify whether any image-analysis software was used.
Results.
L 210-214, 231-238. Did the authors test these percentages for significant differences?
Discussion.
L 250-259. The authors’ statement that “in the Salmoninae family, the most common mode of gonad development is differentiated gonochorism … In contrast, representatives of the Thymallinae subfamily display undifferentiated gonochorism” requires clarification. The authors should define both terms more precisely. They should indicate the precise developmental stages at which grayling diverge from other Salmoniformes.
L 264-267. The authors should report the specific enzymes responsible for OHA conversion in grayling.
L 273-305. It would be useful to summarize these findings in a comparative table listing species, dose, administration route, exposure duration, masculinization rate, intersex incidence, and fertility outcomes.
L 309-311. The authors should define the dose thresholds for sterilization versus intersex induction in grayling (or nearest model).
Round 2
Reviewer 1 Report
Comments and Suggestions for Authors
The author has made point-to-point revisions according to the reviewer's suggestions, and the quality of the manuscript has significantly improved compared to before. I agree to publish according to the journal format requirements.
Reviewer 2 Report
Comments and Suggestions for Authors
No further comment.